# Importance of Diversity, Equity, and Inclusion in the Hepatopancreatobiliary Workforce

**DOI:** 10.3390/cancers16020326

**Published:** 2024-01-11

**Authors:** Timothy A. Rengers, Susanne G. Warner

**Affiliations:** 1Mayo Clinic Alix School of Medicine, Rochester, MN 55905, USA; rengers.timothy@mayo.edu; 2Mayo Clinic Division of Hepatobiliary and Pancreas Surgery, Rochester, MN 55905, USA

**Keywords:** diversity, equity, inclusion, hepatopancreatobiliary, HPB, DEI, surgery

## Abstract

**Simple Summary:**

This review examines the available data that have been reported on diversity within the field of hepatobiliary and pancreas surgery in the United States. The authors review the barriers surmounted by successful hepatobiliary and pancreas surgeons from backgrounds that are historically underrepresented in medicine. The authors found that barriers exist at each point of training: starting from admission into medical school, continuing into surgical residency programs and surgical fellowship programs, and into surgical practice. A pattern emerged that revealed the attrition of underrepresented students at each progression toward hepatobiliary and pancreas surgical practice. The authors further review challenges inherent to workforce homogeneity and introduce evidence-based solutions that promote the achievement of excellence via diversity, equity, and inclusion in hepatopancreatobiliary surgery and beyond.

**Abstract:**

Diversity is a catalyst for progress that prevents institutional stagnation and, by extension, averts descent to mediocrity. This review focuses on the available data concerning hepatopancreatobiliary (HPB) surgical workforce demographics and identifies evidence-based strategies that may enhance justice, equity, diversity, and inclusion for HPB surgeons and their patients. We report that the current United States HPB surgical workforce does not reflect the population it serves. We review data describing disparity-perpetuating hurdles confronting physicians from minority groups underrepresented in medicine at each stage of training. We further examine evidence showing widespread racial and socioeconomic disparities in HPB surgical care and review the effects of workforce diversity and physician–patient demographic concordance on healthcare outcomes. Evidence-based mitigators of structural racism and segregation are reviewed, including tailored interventions that can address social determinants of health toward the achievement of true excellence in HPB surgical care. Lastly, select evidence-based data driving surgical workforce solutions are reviewed, including intentional compensation plans, mentorship, and sponsorship.

## 1. Introduction: Defining the Problem

Structural barriers are often invisible to those in privileged positions but have far-reaching consequences across all aspects of life. In 2008, the World Health Organization Commission on Social Determinants of Health shed some light on these often unseen disparities with a report suggesting that economic and social conditions may influence health status [1]. Inequitable economic, political, social, and psychological processes diminish the physical and mental health of members from marginalized groups [2]. A workforce composed of employees from diverse racial, ethnic, gender, and sexual orientation backgrounds can help combat health disparities [3]. The literature pertaining to workforce diversity in hepatopancreatobiliary (HPB) surgery is sparse. This review aims to not only evaluate the existing data concerning diversity, equity, and inclusion (DEI) in the HPB surgical workforce but also to determine implications for HPB patient care and offer evidence-based solutions.

### 1.1. Characterizing the Problem—Recruitment and Retention of a Diverse HPB Workforce

The demographics of the physician workforce in the United States do not reflect those of our increasingly diverse general population. This is especially pronounced in leadership roles and faculty positions. For instance, medical school faculty from minority groups underrepresented in medicine (URiM) comprise 3.6% Black; 3.3% Hispanic, or Latinx; and 0.1% American Indian and Alaskan Native (AIAN) physicians despite making up 13.4%, 18.5%, and 1.3% of the United States (US) population, respectively [4]. Acknowledgment and attempts at the institutionalized correction of URiM gaps were implemented in 2009 by the Liaison Committee of Medical Education (LCME) as part of an Association of American Medical Colleges (AAMC) initiative [5]. The Accreditation Council for Graduate Medical Education (ACGME) also recently implemented diversity guidelines for residency and fellowship applications. However, a 2020 study conducted by Jarman and Whiting et al. revealed that, despite these guidelines, there was no significant increase in the selection of URiM applicants for interviews for the 2018–2019 application cycle. Interestingly, the study did find a notably higher number of URiM residents at institutions with a higher percentage of URiM faculty [6].

Challenges in recruiting and retaining people of color persist during surgical residency and beyond. Demographic data released by the National Resident Matching Program (NRMP) showed that 73.5% of White applicants matched into general surgery in 2022 versus 64.2% of Black applicants and 57.9% of Hispanic or Latinx applicants [7]. This sets up continued underrepresentation in surgical fellowships that train HPB surgeons. For instance, Complex General Surgical Oncology (CGSO), a fellowship training pathway with an estimated 80% of graduates reporting intent to practice HPB surgery [8], is one of the most competitive general surgery fellowship pathways [9]. A 2023 paper by Collins and Clarke et al. highlights the drop-off of URiM at every progressive stage in training [10]. Collins and Clarke et al. further found that the proportion of White CGSO fellows increased significantly from 54.5% in 2015 to 69.2% in 2020 (*p* = 0.009), with no change in the proportion of Black (5% in 2015 to 2.5% in 2020, *p* = 0.060) and Hispanic or Latinx (4% in 2015 to 4.2% in 2020, *p* = 0.707) fellows [10].

Two additional training routes that can lead to elective HPB surgical practice include the Transplant Accreditation and Certification Council-accredited American Society of Transplant Surgeons (ASTS) transplant surgery fellowships and the Fellowship Council-accredited Americas Hepato-Pancreato-Biliary Association (AHPBA) HPB surgery fellowships. The demographics of URiM ASTS and AHPBA trainees follow a similar pattern to CGSO, with matriculating fellows less representative of the general population than the already low proportions of URiM general surgical trainees [11,12]. This induces ripple effects in practicing HPB surgeon demographics. For instance, the proportion of White transplant surgeons increased by 35% compared with Non-White transplant surgeons from 2000 to 2013 [11]. Additionally, the overwhelming majority of the HPB surgical workforce is male. In the 2019–2020 academic year, 42% of general surgery residents were female yet from 2015–2020, only 22.0% of HPB fellows were female [13]. When we examine senior membership trends in HPB surgical societies like the AHPBA, female representation rivals that of urology and orthopedics [14]. This dearth of diversity is not unique to HPB surgery and is perpetuated throughout academic surgery, broadly. In 2021, Zhu et al. performed a 12-year retrospective cross-sectional analysis of data published by the AAMC. They reported that 69.9% of academic surgeons were White and 74.5% of academic surgeons were male [15]. Academic surgical leadership positions are even less diverse, with 77.7% of full professors and 77.4% of chairs being White [16]. Accordingly, a 2022 strengths, weaknesses, opportunities, and threats (SWOT) analysis commissioned by the AHPBA revealed that a majority of AHPBA-certified fellowship graduates felt that society and fellowship training programs could do more to support URiM and female surgeons [12].

While characterizing precise HPB workforce numbers and demographics is challenging [17], there are data that demonstrate that fewer URiM have the opportunity to join that workforce. For example, URiM medical students, as well as those from low-income backgrounds or under-resourced neighborhoods, were more likely to experience medical school attrition [18]. Moreover, a retrospective cohort study using publicly available data from the AAMC [19] examined 2014–2016 medical school matriculants applying for residency from 2018 to 2021. Among the 37,485 residency applicants, Black and Hispanic male students and AIAN and Hawaiian Native or Pacific Islander female students had the highest rates of unsuccessful graduate medical education (GME) placement. Even when the model was adjusted for sex, race/ethnicity, and United States Medical Licensing Examination (USMLE) Step 2 scores, Black and Hispanic students were less likely to match than White male students.

Once underrepresented trainees match in residency, the attrition threat persists. Haruno and Poon et al. collected AAMC data from 2001 to 2018 pertaining to trainees in surgical residency programs to assess the racial and gender differences in attrition during residency training [20]. The results of their analysis showed that, in this time period, women and URiM trainees were at significantly higher relative risk for attrition than their male and White counterparts, respectively. In this study, attrition was categorized by withdrawal/dismissal due to career change (leaving medicine), health/family reasons, visa issues, military obligations, and involuntary termination. Additionally, they assessed transfer to another specialty. Unintentional attrition was defined by the authors as encompassing all withdrawals, dismissals, and transfers (except for changing careers). The highest attrition (10.6%) and unintended attrition (5.2%) were in Black residents, which is an overrepresentation when compared with the total cohort’s attrition (6.9%) and unintended attrition (2.3%). White residents had lower attrition and unintended attrition than both Black residents and the total cohort at 6.2% and 1.8%, respectively. Black residents were disproportionately at risk of attrition (RR, 1.66; 95% CI, 1.53–1.80; *p* < 0.001) and unintended attrition (RR, 2.59; 95% CI, 2.31–2.90; *p* < 0.001) compared with the total cohort. These data together demonstrate that despite efforts to increase URiM matriculation into medical schools, there are ongoing problems with the pipeline into HPB surgery with fewer URiM students matching into residency and persistent programmatic failure to retain URiM surgical residents.

When designing strategies to increase diversity, it is critical to note that the most effective diversity initiatives target all marginalized groups. For example, a recent study by Iwai and Fayanju et al. analyzed the trends in intersectional demographics among medical students, general surgery residents, and surgical faculty in the United States from 2011 to 2020 [21]. The findings revealed that medical students were more diverse in terms of race/ethnicity and gender compared with surgical faculty. Additionally, there were no significant changes in the representation of URiM individuals among surgical faculty over the study period, while diversity increased among Non-White, non-URiM male medical students; Non-White, non-URiM female residents; and both White and Non-White, non-URiM female faculty. These data suggested that the diversification was primarily among Asian, multiracial, and non-citizen permanent US residents. This study found that faculty gender parity had no correlation with URiM student enrollment. Instead, schools with a higher number of URiM faculty members, particularly male URiM faculty, showed a positive correlation with URiM medical student recruitment and retention.

We cannot fix what we cannot see—and we cannot fully understand the breadth of this problem until we are regularly accounting for our interviewee and trainee demographics in the same way that we should be tracking our surgical outcomes. This was apparent in a study conducted by Lund et al. on HPB applicants at a high-volume North American center. The study design highlights one of the challenges in studying surgical workforce demographics. In this study, creative workarounds were required to identify the genders and races of HPB interviewees and graduates because there were no nationally available data sources regarding either of these demographics for HPB fellowship graduates [22]. While this and other studies do suggest that the surgical workforce is seeing very modest but steady increases in diversity [23], academic surgery has a long way to go before the proportions of practicing URiM surgeons are representative of the US population.

### 1.2. The Importance of Diversity in the HPB Surgery Workforce

Some may ask why it is necessary for a workforce to mirror the demographics of the patient population it serves. A diverse healthcare workforce not only enhances individual patient care outcomes but also contributes to the quality of research by including patients that reflect national demographics. The recognized advantages of greater diversity in the medical field include but are not limited to an increased likelihood of URiM physicians providing superior care for patients of color, working in medically underserved areas, and serving underinsured patients [24,25,26]. Additionally, diversity leads to enhanced research inclusion, as patients from racial and ethnic minority groups are more inclined to participate in clinical trials when a research team member shares their racial or ethnic background, thus facilitating findings that are applicable to a broader population [27,28]. Furthermore, patients may experience improved outcomes when they receive care from female surgeons [29]. Lastly, while the literature is mixed regarding the impact of physician–patient racial concordance on patient outcomes, survey data suggest improved patient experiences with racial concordance [30].

Diversity in the HPB workforce not only fosters innovation and intellectual growth but can also improve workforce fulfillment and safety. For data to understand these phenomena, we must look outside HPB surgeons. Overt discrimination in medical training is known to be associated with decreased productivity and increased rates of substance-use disorder, depression, and suicidality [31]. The impact of subtle comments or actions, often referred to as microaggressions, is a critical and understudied factor in trainee and surgeon professional wellness. Microaggressions communicate derogatory or hostile messages and assumptions based on characteristics such as race, ethnicity, gender, disability, or sexual orientation. This is a well-established phenomenon in the surgical literature [32,33,34], but recent studies have linked these seemingly trivial, often unintentional actions to physician burnout and higher rates of suicidal ideation [35,36].

In a recent study by Anderson et al., an online survey was distributed to medical students across the US to assess their experiences with microaggressions and the impact on mental health and educational satisfaction [37]. The results found a dose-dependent relationship between microaggression and positive depression screening results and decreased medical school satisfaction. This included increased consideration of medical school transfer, withdrawal, and the belief that microaggressions were a normal part of medical school culture among students who reported experiencing at least one microaggression weekly.

Upon completion of training, URiM surgeons can experience intentional and unintentional interpersonal racism from patients, colleagues, and support staff. This somewhat nebulous concept is difficult to study; thus, we must turn to our social science colleagues to comprehend that the impact and severity of discrimination and microaggressions in the workplace can vary significantly between racially diverse environments and predominantly White workplaces. Studies suggest that these experiences tend to carry less harm in diverse settings compared with homogenous ones [38]. For instance, Meyers et al. developed a survey on work experiences with discrimination and microaggressions that was distributed to monoracial people of color, multiracial individuals, and White individuals across the US [38]. Not surprisingly, people of color experience fewer instances of discrimination and microaggressions in racially diverse contexts compared with homogenously White contexts. This suggests that diversity can have positive effects by reducing discrimination experiences, potentially because of either increased contact with racially different individuals or the role of a “majority–minority” context, which can diminish the salience of racial identity for racial minorities. The study also highlights the role of societal racial hierarchies in the daily lived experiences of White individuals who experience no differences in discrimination or microaggressions between diverse and homogenously White workplaces. Thus, while a diverse workforce is not a panacea for preventing microaggressions, a diverse workplace may at least offer modest defenses against them to avert URiM burnout and attrition.

Many discussions surrounding the benefits of diversity are intertwined with moral and ethical considerations, asserting that diversity is inherently the “right” course of action. Individuals in positions of privilege or belonging to the majority often argue that diversity in medical education carries a cost, with a particular focus on what they perceive as a shortage of qualified URiM applicants [39]. In contrast, social science researcher Scott Page, Ph.D., has authored a book titled The Diversity Bonus [40]. Drawing on evidence that diversity of thought and experience can actually enhance productivity and profitability on corporate teams, Page concludes that diversity actually enhances a system and should be seen as an asset that provides financial, creative, and academic advantages.

In his book, Page presents the idea of diversity as having a diverse set of problem-solving approaches or “tools” that can be more effective than relying on a single approach. He argues that different individuals bring unique perspectives, skills, and problem-solving methods to the table. Just as a toolbox with a variety of tools can enable the completion of different tasks, a diverse group of people with a range of problem-solving approaches can be more effective in tackling complex challenges (Figure 1).

Diversity of thought and experience can also extend to country of origin. A study conducted by Baker and Jeyarajah et al. in 2016 demonstrated that, after interviews, curriculum vitae, and letters of recommendation, training at a US or Canadian residency was one of the most important factors in evaluating an applicant for North American HPB fellowship training [41]. However, international medical graduates (IMGs) have historically faced much more adversity in matching to a US residency program [42,43]. While there are many complexities to this discussion, it is worth noting that our IMG colleagues contribute greatly to their fields—including HPB surgery—and we should acknowledge discrimination against IMGs in training selection, referral patterns, society leadership, and promotion if we are to have a complete discussion about diversity in HPB surgery.

A diverse workforce can both create a safe and financially favorable place of employment and contribute to improved patient care. Considering the well-documented disparities in HPB cancer care, the field must take the current paucity of URiM HPB trainees, faculty, and senior leadership seriously. Despite advances in cancer surveillance and earlier intervention, racial disparities in the treatment and outcomes of patients with HPB malignancies persist. For instance, Black individuals have pancreatic tumors resected at lower rates and present at a later stage [44]. Similarly, Hispanic and African-American race are associated with decreased rates of transplantation in hepatocellular carcinoma [45]. While longstanding structural barriers to care and socioeconomic status [46] likely drive some of these disparities, patients from underrepresented groups are very likely to experience racial bias even when they are members of higher income quartiles [2]. Here, again, a diverse workforce has been shown to help patients from underrepresented groups see and be seen by medical providers.

### 1.3. Surgical Referral Patterns

Given sparse data, we must again look beyond HPB surgery to understand the costs of bias toward both patients and physicians. Landon et al. recently published an observational study on patterns of patient sharing between primary care physicians (PCPs) and six of the most frequently referred-to specialties and found that PCPs shared Black patients with fewer specialties relative to their White patients [47]. This study is important because it provides real-world evidence of how racial bias can contribute to restricted healthcare access. HPB was not a specialty specifically examined in the Landon et al. study, but a recent retrospective cohort study by Yilma et al. investigated the factors associated with liver transplant referral and found that Non-Hispanic Black patients had lower odds of referral [48]. Hollingsworth et al. identified significant provider care team segregation between White and Black patients undergoing coronary artery bypass graft within US hospitals, with higher segregation associated with higher operative mortality rates for Black patients [49]. Patients may encounter “invisible” barriers within their healthcare environment that hinder access to experienced, high-volume surgeons, potentially affecting their outcomes. In some cases, patients, particularly among Black communities, may choose lower-volume hospitals even when high-volume options exist in their communities because of network affiliations [50]. It remains uncertain whether individual physicians are aware of these network dynamics and their impact on patient outcomes or what actions they would take if they possessed this knowledge. Taken together, these data make it increasingly clear that without intentional, concerted efforts to see, measure, and mitigate deficits in diversity, equity, and inclusion, we will continue to perpetuate injustices for our patients and colleagues.

In addition to bias toward patients, referral patterns are influenced by bias toward surgeons as well. Several studies have documented gender bias in referral patterns. For instance, Dossa et al. found that male physicians have strong preferences to refer to other male surgeons, whereas female physicians are less influenced by surgeon sex but still refer preferentially to male surgeons; moreover, female surgeons more commonly receive non-operative referrals [51]. These findings held true even when accounting for surgeon characteristics such as age, availability, experience, generational differences, and patient characteristics. Furthermore, in an elegant thesis, Dr. Heather Sarsons found that female surgeons suffer greater professional consequences following a patient death, whereas referrals to male surgeons actually increase after major complications or death [52]. A 2019 survey distributed to members of the Asian-Pacific Hepato-Pancreato-Biliary Association (APHPBA) points to a lack of female role models, family or childcare issues, and gender discrimination as reasons that trainees may not pursue a career in HPB surgery [53]. Taken together, these findings suggest that systemic bias may underpin these referral patterns and the key reasons that female-identifying surgeons are not choosing HPB as a career.

In summary, the barriers contributing to disparities in the healthcare workforce are pervasive, but based on the available data, they appear more pronounced within HPB surgery than with other sub-specialties. These disparities have profound consequences, exacerbating inequities in HPB patient care and outcomes. Embracing diversity in healthcare is not just a moral imperative but also a practical advantage, offering a broader problem-solving toolbox and improved patient care. Physicians must rely on their training in treating problems with evidence-based solutions to cure the disease of inequity: identify the etiology of symptoms and treat with researched resolutions and tactics.

## 2. Discussion: Identifying Solutions

The ongoing disproportionately low numbers of URiM surgical residents and the even lower numbers of URiM HPB fellows and surgeons invite a troubling vicious cycle. The small number of senior URiM HPB surgeons is taxed with mentoring more than their share of prospective HPB surgeons, further burdening their own time and perpetuating the risk of burnout. The decreased availability of mentorship/role models alongside microaggressions against choosing HPB surgery discourages a significant number of surgical trainees from choosing the field. This lack of representation may lead to an increased risk of overt and/or subtle discrimination and compromises workforce safety for URiM surgeons [31,32,33,34,38]. These systemic issues contribute to higher rates of substance-use disorder, depression, and suicidality among URiM surgeons [35,36]. Addressing these challenges is paramount to generating evidence-based solutions to promote greater diversity and equity among the HPB surgery workforce and, ultimately, to provide better care for the diverse patient populations it serves.

### 2.1. Evidence-Based Solutions: Workforce

When considering the rate of attrition that URiM physicians face when seeking a career in HPB surgery, one of the barriers to successful transition between phases of training pertains to access to quality mentorship and sponsorship. Now that the United States Medical Licensing Examination (USMLE) Step 1 examination no longer has a three-digit score to filter residency applicants, the grades while a student is in their clerkships and the number of research publications authored by the applicant hold much more significance in the application process. However, a recent survey of medical students on their experiences with clerkship grading demonstrates that only 44% of students felt the evaluations were fair, and a majority of students felt that the grading system encourages performance goals rather than fostering a supportive learning environment aimed at improvement [54]. Additionally, a recent publication by Hanson et al. found and characterized racial/ethnic disparities in the grading of medical school clerkship evaluations, which can only be rectified if faculty first identify their own biases and evaluate students without the first-impression gestalt [55].

The shift to a pass/fail model for standardized exams in academia places increased importance on research productivity [56,57], which could exacerbate racial disparities in authorship, particularly among Black and Hispanic physicians, as evidenced by a decline between 1990 and 2020 [58]. Furthermore, bias in support from the National Institute of Health (NIH) may contribute to academic promotion inequities [59]. Peer-review practices in surgery journals outside of HPB may shed light on pervasive problems. For instance, peer-review practices in orthopedic surgery journals are problematic, with only two-thirds using double-blinded peer review and 40% allowing author-suggested reviewers, potentially introducing bias [60]. These findings are consistent with the experience of female-identifying academic surgeons within HPB surgery as well. For instance, women comprised 26% of first authors and only 10% of senior authors in HPB-related manuscripts accepted by top surgical journals in 2018 [61]. Implementing stricter blinding practices in peer review could help boost the productivity of URiM trainees and enhance retention throughout their academic journeys.

Another method that has shown demonstrable benefit toward increasing URiM trainee exposure to mentors across surgical sub-specialties is the creation of URiM travel awards and other grant funding to attend or participate in annual conferences—or to have their curriculum vitae or research reviewed in preparation for fellowship applications [62,63]. In addition to the obvious benefits of connecting URiM trainees with likeminded peers, mentors, and leaders in the field, the resultant influx of URiM conference attendees helps build community and safety for marginalized attendees who might otherwise feel unsafe attending a traditionally less diverse society meeting. Research with objective measures pertaining to mentorship models in academic medical centers is sparse, but a 10-year longitudinal study conducted by Daley et al. may shed light on the effect of intentional mentorship programs. In a cohort of 30 URiM junior faculty, 92% of those eligible for promotion were successfully promoted to associate professor. The key factors contributing to their success included access to senior faculty mentors, peer networking, professional skill development, and an understanding of institutional culture, emphasizing the importance of a comprehensive faculty development program in supporting URiM faculty in academia [64].

Racial and gender pay equity and transparency is another proven way to recruit and retain a more diverse workforce, including an HPB one. While much of the literature on physician compensation is focused on gender [65], a 2023 Medscape survey showed that White physicians, on average, earned 13% more annually than their Black colleagues [66]. Data outside the medical field would suggest that these gaps are much higher for women of color—with Black women earning ~70% and Hispanic women earning ~65% as much as their White male colleagues [67]. One way that employers can be true to their commitments to DEI is through transparent, structured compensation plans (Figure 2). Options are myriad but include salary-based compensation models wherein physicians are paid the same within a specialty regardless of individual productivity, billing, or collections. Bonuses are not awarded for performance, but rather, increased compensation comes from assuming additional responsibility (i.e., becoming a division chair). Morris et al. reported on a single-center academic department of surgery and found that a structured compensation plan improved the gender pay gap within their department [68]. A recent analysis by Hayes et al. demonstrated 96% pay equity when the Mayo Clinic, a multisite academic hospital system, committed to structured compensation plans for their physicians [69]. Because few surgical departments have historically strived for or achieved pay equity, even fewer can study its effects on recruitment and retention. At present, existing evidence connects the attrition of surgical faculty predominantly to disparities in pay and promotion [70]; by extrapolating from this link, it is plausible to infer that fostering a more transparent and equitable approach toward achieving pay and promotion parity may effectively retain surgeons from marginalized groups.

### 2.2. Evidence-Based Solutions: Patient Care

There are few studies demonstrating successful interventions to enhance HPB care for at-risk patients from marginalized groups. Considering the multifactorial nature of the structural barriers to successful care, some studies recommend that surgeons screen for social determinants of health in the clinical setting and offer the appropriate resources to address their patients’ needs [46]. This, of course, is much easier said than done. One consistently successful intervention is the incorporation of patient navigators and community health workers or volunteers. Patient navigation and community health worker interventions aim to address social determinants of health and have consistently demonstrated their effectiveness in reducing health disparities [71,72,73,74]. These interventions provide individualized support to cancer patients and their families, helping them overcome healthcare system barriers, ensuring timely access to high-quality healthcare, and offering culturally competent education and counseling. Studies have shown that both patient navigation and community health worker interventions increase cancer screening rates and reduce disparities, particularly among medically underserved populations [71,75]. At Grady Memorial Hospital, one such model demonstrated improved treatment adherence by recruiting patient care navigators. These navigators assisted in coordinating radiology and liver clinic appointments, managing referrals, ensuring appointment reminders, and successfully reengaging 55% of patients who missed exams, resulting in the completion of subsequent ones. Moreover, the implementation of patient navigators effectively guided high-risk patients through hepatocellular carcinoma screening and subsequent treatment [76].

A recent study by Griesemer et al. explored the Accountability for Cancer Care through Undoing Racism and Equity (ACCURE) intervention, which successfully mitigated racial disparities in cancer treatment completion at two U.S. cancer centers [77]. The ACCURE intervention encompassed multiple approaches, such as educating healthcare providers about the underlying factors contributing to healthcare disparities [78]. It also featured a Real-Time Registry data system that monitored patient progress in real time, with data broken down by patient race [79]. Additionally, nurse navigators were enlisted to facilitate patient engagement by utilizing data-driven care [80]. The research team identified transparency and accountability as key mechanisms of change within ACCURE, achieved through specific components such as real-time quality metric reporting by patient race and nurse navigators trained in anti-racism. In a trial comparing early-stage breast or lung cancer interventions with controls, significant disparities in treatment completion between Black and White patients were evident in retrospective and concurrent controls (Black: 79.8% vs. White: 87.3%, *p* < 0.001; Black: 83.1% vs. White: 90.1%, *p* < 0.001). However, within the intervention group, the Black–White completion disparity lessened (Black: 88.4% vs. White: 89.5%, *p* = 0.77), with Black patient completion in the intervention comparable to or better than White patients in retrospective (OR 1.6; 95% CI 0.90–2.9) and concurrent (OR 1.1; 95% CI 0.59–2.0) controls [79].

## 3. Conclusions

Achieving diversity, equity, and inclusion in HPB surgery is an ongoing challenge that should concern all HPB surgeons. With obstacles looming at each step from medical school through residency, fellowship, and continuing into practice, URiM physicians often face significant adversity just to enter and remain on track for HPB careers. Additionally, a diversity deficit in HPB surgery likely contributes to the perpetuation of significant disparities in HPB surgical care delivery and patient outcomes. This review of the literature suggests that diversity not only enhances patient care and research but also creates a more supportive work environment, reducing the risk of physician burnout and discrimination-related consequences. To promote workforce equity, implementing evidence-based solutions such as equitable compensation plans, improved mentorship opportunities, blinding practices in peer review, and addressing biases in research funding is crucial. To reduce health disparities for our patients, the recruitment of patient navigators should be prioritized in HPB cancer care. Achieving DEI in HPB surgery is not only a moral imperative but also a practical advantage for enhancing patient care, research quality, and workforce satisfaction, necessitating a comprehensive approach across medical education, training, academic, and research environments.

## Figures and Tables

**Figure 1 cancers-16-00326-f001:**
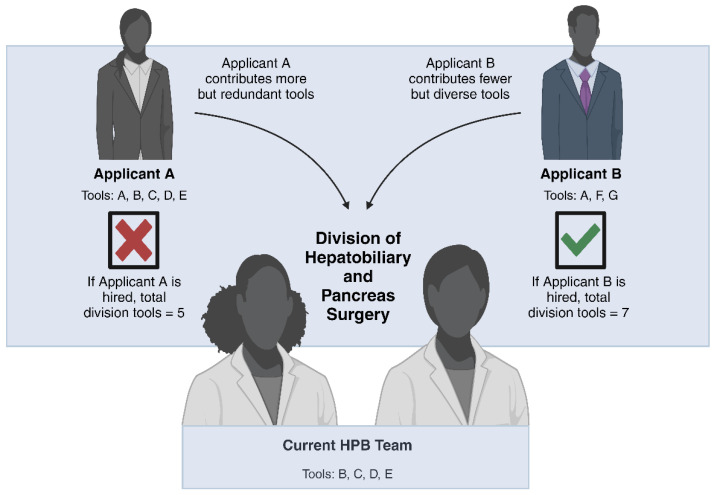
Concept adapted from The Diversity Bonus by Scott Page, Ph.D. Each clinical or surgical skill is represented by a tool labeled with a letter (A–G). Applicant A is the more skilled candidate with 5 skills, while Applicant B only has 3 skills. While they are both able to fill a gap in the current practice by offering Tool A, the best option is Applicant B because they offer a unique set of tools to diversify the current range of available skills within the Division of Hepatobiliary and Pancreas Surgery. Legend: HPB: hepatopancreatobiliary.

**Figure 2 cancers-16-00326-f002:**
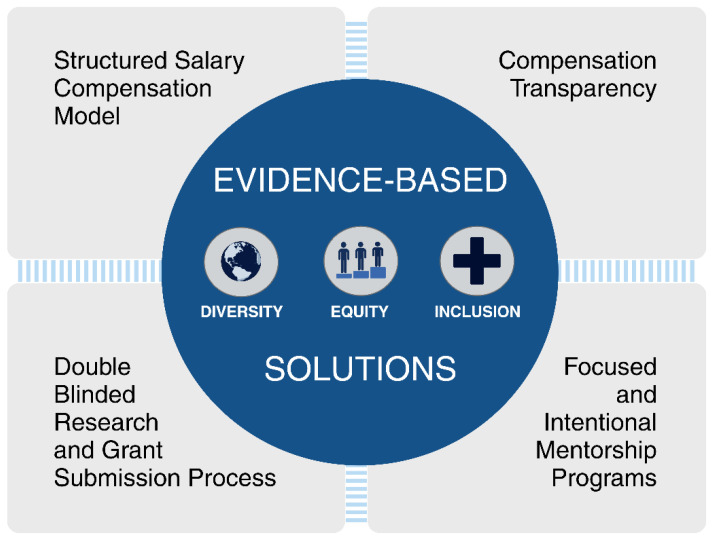
Evidence-based solutions to increasing diversity in the hepatopancreatobiliary surgical workforce.

## Data Availability

No new data were created or analyzed in this study. Data sharing is not applicable to this article.

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
