# Peer review of "Importance of Diversity, Equity, and Inclusion in the Hepatopancreatobiliary Workforce"

_cancers, 2024, doi:10.3390/cancers16020326_

Round 1

Reviewer 1 Report

Comments and Suggestions for Authors

I would like to congratulate Authors on their work. This paper is the comprehensive contrubution to the very important topic. It is very universal problem accompanied by the huge emotional burden. Many doctors from groups URiM suffer from different types of discrimination and there is an urgent need to change this situtation, what translates into benefits to the society as a whole.  Despite Authors refer to the HPB field, I believe that the take home message could be spread to other discplines and in a more general sense to our everyday life. Thus I am very happy that Authors made this effort to share such awareness with onco-surgical community.  For sure it is an important step towards gradual improvement which we are hoping for. 

Author Response

Thank you for your thoughtful review and feedback. The authors agree that the take home message that improving diversity has far reaching benefits for HPB surgery and patients also applies to disciplines beyond HPB. 

Reviewer 2 Report

Comments and Suggestions for Authors

A very interesting paper . The same issues should be investigated in every country.

Author Response

Thank you for reviewing our work. Our focus was on the United States workforce due to the heterogeneity in national demographics but the authors agree that more research must be done to investigate the issue of diversity internationally.

Reviewer 3 Report

Comments and Suggestions for Authors

This was well formated article pertaining the diversity, equity and inclusion in HPB workforce. In conclusion, evidence-based data tell us the surgical workforce solutions should started from intentional compensation plans, mentorship, and  sponsorship. However, I strongly suggested this article might match the readers in the HPB  or Surgery specific journals instead of Cancer specified journal.

Author Response

Thank you for the attentive read through our manuscript. This summary accurately portrays the recommended solutions toward increasing diversity within the field. Although the authors understand your concern about the readership of Cancers, this manuscript has been submitted for consideration of inclusion in a special issue of the journal titled, “Innovation in Surgical Treatment of Hepato Pancreatico Biliary (HPB) Cancers.” Therefore, we believe that in the context of this special issue, our work has a unique position to highlight the disparities in the HPB surgical workforce.

Round 2

Reviewer 3 Report

Comments and Suggestions for Authors

This is a well preparing manuscript, but extremely sorry I still insisted my comment as that of previous.